# Eosinopenia as a Prognostic Biomarker for Noninvasive Ventilation Use in COPD Exacerbations

**DOI:** 10.3390/jpm13040686

**Published:** 2023-04-19

**Authors:** Konstantinos Bartziokas, Evgenia Papathanasiou, Andriana I. Papaioannou, Ilias Papanikolaou, Emmanouil Antonakis, Ioanna Makou, Georgios Hillas, Theodoros Karampitsakos, Ourania Papaioannou, Katerina Dimakou, Vasiliki Apollonatou, Galateia Verykokou, Spyros Papiris, Petros Bakakos, Stelios Loukides, Konstantinos Kostikas

**Affiliations:** 1Respiratory Medicine Department, University of Ioannina, 45110 Ioannina, Greece; 22nd Respiratory Medicine Department, National and Kapodistrian University of Athens, 10679 Athens, Greece; 3Respiratory Medicine Department, Corfu General Hospital, 49100 Corfu, Greece; 45th Respiratory Medicine Department, Sotiria Chest Hospital, 11527 Athens, Greece; 51st Respiratory Medicine Department, National and Kapodistrian University of Athens, 10676 Athens, Greece

**Keywords:** COPD, exacerbations, eosinophils, hospital admission, non-invasive ventilation

## Abstract

Background: In recent years, blood eosinophils have been evaluated as a surrogate biomarker for eosinophilic airway inflammation and as a prognostic indicator of the outcomes of hospitalized COPD subjects. During an exacerbation of COPD, eosinopenia has been proposed as a prognostic marker of adverse outcomes. Objectives: The aim of the present post hoc analysis was to elucidate the effectiveness of blood eosinophils for predicting the need of NIV in subjects with COPD exacerbation. Methods: Consecutive subjects admitted to a hospital for COPD exacerbation were included in the analysis. The eosinophil count from the first complete blood count was used to designate the eosinophil groups. The relationship between the clinical characteristics and blood eosinophil counts, as dichotomized using 150 cells/μL, was evaluated. Results Subjects with blood eosinophil number < 150 k/μL had a more severe disease on admission compared to subjects with ≥150 k/μL, regarding pH 7.400 (7.36, 7.44) vs. 7.42 (7.38, 7.45), *p* = 0.008, PO_2_/FiO_2_ levels 238.1 (189.8, 278.6) vs. 276.2 (238.2, 305.6), *p* < 0.001, CRP (mg/L) levels 7.3 (3.1, 19.9) vs. 3.5 (0.7, 7.8), *p* < 0.001 and required a longer hospital stay (days) 10.0 (8.0, 14.0) vs. 5.0 (3.0, 7.0) *p* < 0.001 respectively. The number of blood eosinophils correlated with the levels of CRP upon admission (*p* < 0.001, r = −0.334), with arterial pH upon admission (*p* < 0.030, r = 0.121), with PO_2_/FiO_2_ (*p* < 0.001, r = −0.248), and with duration of hospital stay (*p* < 0.001, r = −0.589). In the multinomial logistic regression analysis, blood eosinophil count < 150 k/μL was an independent predictor of the use of NIV during hospital stay. Conclusion: During COPD exacerbation, low blood eosinophil levels upon admission are related to more severe disease and can be used as a predictor of the need of NIV. Further prospective studies are needed to identify the use of blood eosinophil levels as a predictor of unfavorable outcomes.

## 1. Introduction

Exacerbations of chronic obstructive pulmonary disease (COPD) are events characterized by increased symptoms (mainly dyspnea cough and sputum), which are usually accompanied by tachypnea and tachycardia and are associated with increased lung and systemic inflammation [1]. COPD exacerbations, especially more severe ones, that require hospital admission often lead to adverse events and are associated with increased mortality risk [2]. COPD exacerbations might be caused by different factors, such as bacterial or viral infections or by increased air pollutants [1]. However, the identification of the specific factor that causes the exacerbation is difficult; thus, in most cases, the treatment is empirical according to the patient’s clinical presentation and laboratory findings. 

Eosinophilic airway inflammation accounts for approximately 25% of patients admitted to the hospital for a severe COPD exacerbation [3]. In recent years, blood eosinophils have been evaluated as a surrogate biomarker for eosinophilic airway inflammation and as a prognostic indicator of the outcomes of hospitalized COPD subjects [4,5]. During exacerbations of chronic obstructive pulmonary disease, eosinopenia has been proposed as a prognostic marker of adverse outcomes [6], and blood eosinophils have been used as a helpful biomarker for guiding treatment decisions in COPD subjects hospitalized due to disease exacerbation [7].

Noninvasive ventilation (NIV) has become part of the standard intervention for subjects with COPD exacerbation [8] and provides several advantages, including shorter duration of stay in the intensive care unit (ICU), shorter hospital stay, and fewer complications [9]; it has also been recognized as an intervention that decreases mortality in these patients [1]. More than half of COPD exacerbations are triggered by bacterial and/or viral infections, which can aggravate chronic inflammatory responses [10]. It was previously reported that among subjects with a COPD exacerbation, those in whom NIV was unsuccessful presented a higher level of serum inflammatory markers (PCT and CRP) compared to subjects in whom NIV was applied successfully [11]. Hence, it is advisable to enhance the accuracy of early identification of subjects with COPD exacerbation who may need the application of NIV. Data from prospective studies have evaluated the predictive role of blood eosinophils in NIV of hospitalized COPD subjects [4,5]. According to the abovementioned findings, the objective of this post hoc analysis was to elucidate the effectiveness of blood eosinophils for predicting the need of NIV in COPD subjects hospitalized for a severe COPD exacerbation. 

## 2. Methods

### 2.1. Study Design

This study is a post hoc analysis of previously published data [12]. The detailed study design has been previously published [12]. Briefly, we enrolled consecutive COPD subjects admitted to a hospital for acute exacerbation. The data were collected from four Respiratory Medicine departments in tertiary referral hospitals in Greece between November 2015 and January 2018. All subjects had a previous diagnosis of COPD confirmed by a respiratory physician. Subjects with a history of respiratory disorders other than COPD, those with inability or unwillingness to cooperate with the investigators, and those without available spirometry data were excluded. Subjects were also excluded if their admission was related to a cardiac event, including heart failure, unstable angina and acute coronary syndrome, as confirmed by a cardiologist. Finally, we did not include subjects with chronic comorbidities or other conditions that might significantly influence blood eosinophil levels (i.e., parasitic and fungal diseases, allergic reactions, adrenal conditions, skin disorders, toxins, autoimmune or endocrine disorders tumors, and use of systemic corticosteroids prior to admission). The study protocol was approved by the local ethics committee (501/17-07-2018 and 9158/18-07-2013), and all participants provided written informed consent.

Upon admission, patient demographics were recorded, including age, sex, body mass index (BMI), smoking habit, prior pharmacological treatment, and comorbidities. Clinical parameters, including vital signs, level of dyspnea (according to the Borg scale), and results from arterial blood gas analysis, were also recorded. The Charlson comorbidity index score [13] was calculated according to each patient’s medical history and personal medical records. Blood samples were collected from each patient upon admission to the emergency department and prior to the initiation of any inhaled or systematic treatment, and laboratory results, including total blood count, complete blood count, C-reactive protein (CRP) and basic biochemistry results, were also recorded. The diagnosis and classification of airflow limitation was based on post-bronchodilator spirometry upon stable condition (usually performed 4–8 weeks after the patient’s discharge), according to the GOLD guidelines. Furthermore, the pharmacologic treatment of all patients (including administration of bronchodilators, antibiotics and systemic corticosteroids) was administered according to the GOLD recommendations [1]. Finally, all patients who underwent treatment with NIV suffered from respiratory acidosis. 

### 2.2. Identification of Eosinophilic Phenotype

The eosinophil count from the first complete blood count obtained in hospital (usually in the emergency department) was recorded and was used to determine the number of blood eosinophils. Blood sample collection and measurement of blood eosinophils were performed upon admission and before the administration of any treatment. We determined the relationship between the clinical characteristics and blood eosinophil counts, as dichotomized using 150 cells/μL blood as a cut-off, in subjects with COPD exacerbation. We selected this cut-off as a value since, in the initial study [12], it was shown that COPD patients who were admitted to the hospital for COPD exacerbation and had an eosinophil number greater than 150 cells/μL had milder disease upon admission, required shorter length of hospital stay, and had a lower 30-day and 1-year mortality [12]. Additionally, a previous study has shown that treatment with ICS as per physicians’ decision may be beneficial for exacerbation prevention during follow-up in subjects with ≥150 cells/μL upon admission [14].

### 2.3. Statistical Analysis

Normality of distributions was checked using the Kolmogorov–Smirnov test. Comparisons between groups were performed using chi-squared tests for categorical data, as well as unpaired *t*-tests or Mann–Whitney U-tests for normally distributed or skewed numerical data, respectively. Correlations were performed with Spearman’s rank correlation coefficient. Multinomial logistic regression analysis was performed to detect possible predictors of the use of NIV during hospitalization for COPD exacerbation. *p*-values < 0.05 were considered statistically significant. The data were analyzed using SPSS 24.0 for Windows (SPSS Inc., Chicago, IL, USA).

## 3. Results

### 3.1. Study Participants According to the Number of Blood Eosinophils upon Admission 

A total of 388 subjects (83.5% male and age [median (IQR), 72 (66, 78) years]) were included in the study. For our cohort, 213 (54.9%) subjects had an absolute blood eosinophil number < 150 k/μL. Subjects with blood eosinophil number < 150 k/μL had a more severe disease on admission compared to subjects with ≥150 k/μL, regarding arterial blood pH 7.400 (7.36, 7.44) vs. 7.42 (7.38, 7.45), *p* = 0.008, arterial PO_2_/FiO_2_ levels 238.1 (189.8, 278.6) vs. 276.2 (238.2, 305.6), *p* < 0.001, CRP (mg/L) levels 7.3 (3.1, 19.9) vs. 3.5 (0.7, 7.8), *p* < 0.001, a higher level of dyspnea according to the Borg scale 7.0 (5.0, 8.0) vs. 5.0 (3.0, 6.0), *p* < 0.001 and required a longer hospital stay (days) 10.0 (8.0, 14.0) vs. 5.0 (3.0, 7.0), *p* < 0.001 respectively. Furthermore, the subjects with blood eosinophil number < 150 k/μL upon admission had a greater number and severity of comorbidities (as scored by the Charlson comorbidity index, 2.0 (2.0, 4.0) vs. 2.0 (1.0, 3.0), *p* = 0.004) and required the use of NIV more often during hospital stay compared to the subjects with ≥150 k/μL 61 (28.6%) vs. 30 (17.1%), *p* = 0.008. The results are presented in Table 1.

### 3.2. Correlations between Blood Eosinophil Levels and Clinical and Laboratory Characteristics upon Admission

The number of blood eosinophils correlated with the levels of CRP upon admission (*p* < 0.001, r = −0.334), with arterial pH upon admission (*p* < 0.030, r = 0.121), with PO_2_/FiO_2_ (*p* < 0.001, r = −0.248), and with the duration of hospital stay (*p* < 0.001, r = −0.589). The correlations between blood eosinophil numbers and characteristics of the study participants are presented in Table 2.

### 3.3. Predictors of the Need of Non-Invasive Ventilation

In the multinomial logistic regression analysis, the use of NIV was used as the dependent variable, whereas age, gender, BMI, Charlson comorbidity index score, CRP and level of blood eosinophils < 150 k/μL were used as the independent variables. Blood eosinophil number < 150 k/μL was an independent predictor of the use of NIV during hospital stay. The results of the multinomial logistic regression analysis are presented in Table 3.

## 4. Discussion

In this study, we showed that COPD patients who were admitted to hospital for a COPD exacerbation and had lower numbers of blood eosinophils had a more severe disease upon admission (according to clinical and laboratory characteristics) and required a longer hospital stay, compared to patients with higher numbers of blood eosinophils. Furthermore, low blood eosinophil levels during COPD exacerbation (i.e., <150 cells/μL) was an independent risk factor of the need of NIV during hospitalization.

The association of the levels of blood eosinophils during a COPD exacerbation with different disease outcomes has been described in previous studies [4,5,12], all showing that patients with lower blood eosinophil counts are at a greater risk. Since the severity of an exacerbation is associated with the severity of symptoms and the level of functional and respiratory impairment of patients [1], it is important to find inexpensive and available biomarkers for the early recognition of patients who will probably need closer monitoring or who will probably require early admission in intensive care units. Our study shows that the number of blood eosinophils might be used as such a biomarker since it is able to predict which patients have greater possibility of deteriorating during their hospital stay and provide the opportunity for the treating physician to offer them closer monitoring and more intensive treatment. 

Interestingly, in our study, we observed that the group of patients with lower blood eosinophil levels was also characterized by higher CRP levels. CRP is well known as a biomarker associated with exacerbations mainly caused by bacterial infections, and this association has good sensitivity and specificity [1]. Furthermore, it has been reported that COPD exacerbations caused by bacterial infections are associated with poor clinical outcomes [14,15]. In our study, hospitalized COPD subjects with low eosinophil levels upon admission (<150 cells/μ) required a longer length of hospital stay compared to patients with higher blood eosinophils, and this observation might be related to a more severe exacerbation and/or a slower response to treatment. 

The need for NIV use in COPD patients hospitalized for acute exacerbations has been reported to be closely related to the underlying condition (e.g., disease severity, older age, frailty, muscle mass or comorbidities, such as heart failure or sleep apnea syndrome) [3]. However, in our study, the Charlson comorbidity index score, although different between patients with different levels of blood eosinophils (higher in the group of patients with a lower blood eosinophil count), was not an independent predictor of the need of NIV during hospital stay in exacerbating COPD patients. This fact leads to the hypothesis that a low blood eosinophil count reflects a more severe inflammatory process directly related to the respiratory impairment of patients during the COPD exacerbation, which seems to be irrespective of the general health status according to the presence or absence of other comorbid diseases. 

Although weak, the correlations of the number of blood eosinophils with factors of disease severity, such as arterial blood gases, CRP and dyspnea, show a possible pathophysiological connection between the type of inflammation and clinical deterioration during COPD exacerbations. Higher numbers of blood eosinophils correlate with less severe disease in respect to symptoms and inflammatory markers and with earlier hospital discharge, showing that these patients probably have a better response to therapy, which basically includes bronchodilators and systemic corticosteroids. 

An interesting finding in our study was that respiratory dysfunction, as expressed by PaO_2_/FiO_2_ (mmHg) and partially by arterial pH, differed among the groups with different blood eosinophil levels, showing again that subjects with lower blood eosinophil counts were characterized by more severe COPD exacerbation. Our findings are in agreement with a previous study [5], in which subjects with COPD exacerbation and low blood eosinophil numbers were characterized by higher levels of inflammatory biomarkers (leucocytes, neutrophils and lactate dehydrogenase levels) compared to subjects with higher blood eosinophil counts. In this study, a low blood eosinophil count was also an independent risk factor of the need for the use of NIV, a result that is also in accordance with our study.

Interestingly, in our study, the median pH value and the interquartile range (IQR) were normal in both groups, although there was a statistically significant difference between the two groups. However, we must point out that these values were recorded upon admission and the aim of our study was to determine the prognostic value of blood eosinophils on the need of NIV during hospital stay and not upon admission. This fact makes our observation even more important since it helps the treating physician to recognize early on patients who are at greater risk for a poorer outcome and offer them more intensive treatment and closer monitoring. 

Low blood eosinophil levels have already been reported as a risk factor associated with the need for a longer duration of hospital stay and a higher 30-day and all-cause 1-year mortality [12], while they have also been identified as a reliable marker of admission to medical intensive care units in subjects with sepsis [16]. All aforementioned outcomes are probably associated with the severity of exacerbation, and, therefore, we speculate that low blood eosinophil levels may serve as an indicator of a severe COPD exacerbation. Finally, it is important to mention that our observation for the association of low eosinophil levels with more severe disease and worse COPD outcome is also in accordance with the previously described DECAF score (which also includes the presence of “eosinopenia”), which is a multi-component score designed to identify subjects with a higher hospital mortality risk during COPD exacerbation [17].

Our study has some limitations. First, it is a post hoc analysis, which means that it was not designed specifically to examine the predictive value of the number of blood eosinophils on the need of using NIV during hospitalization for severe COPD exacerbation. However, the fact that all data were prospectively collected, and the patients included were admitted in four different Respiratory Medicine departments in tertiary referral hospitals, increases the importance of the study. Secondly, there was no specific prespecified algorithm to direct treatment during hospital stay regarding both pharmacological treatment (antibiotics, bronchodilators and corticosteroids) and non-pharmacological treatment, including the use of NIV in our group of patients. Furthermore, no specific criteria at the time of discharge were provided by the study protocol. Nevertheless, since all data were collected from tertiary referral hospitals in which local hospital algorithms are always in accordance with the GOLD recommendations for both treatment and discharge decisions [1], we can ensure that these patients always received optimal therapy for their COPD exacerbation, that NIV was always administered when there was an indication, and that discharge was always performed according to the GOLD recommendations [1]. Finally, we did not include any data on the number of blood eosinophils upon stable condition, but the fact that we used the values of blood eosinophil upon admission to the emergency department before the administration of any treatment to patients allowed a clear evaluation of the relationship between the number of blood eosinophils during the onset of a COPD exacerbation and the outcome of the patients regarding the need of NIV. 

## 5. Conclusions

In conclusion, our results show that low blood eosinophil levels (<150 cells/μL) upon hospital admission for a COPD exacerbation can be used as a biomarker of more severe disease and as a predictor of the need of NIV during hospital stay. This observation probably can give the treating physician the opportunity to recognize early on which COPD patients are at an increased risk for adverse outcomes in order to be able to offer them closer monitoring and early admission in an intensive care unit to provide an opportunity for a better outcome. However, since our results come from a post hoc analysis, further prospective studies are needed to identify the use of blood eosinophil levels as a predictor of COPD exacerbation severity and probable unfavorable outcomes. 

## Figures and Tables

**Table 1 jpm-13-00686-t001:** Demographic and functional characteristics of the study participants.

Variable	Blood Eosinophils < 150 cells/μL (*n* = 213)	Blood Eosinophils ≥ 150 cells/μL (*n* = 175)	*p*-Value
**Age (years)**	73.0 (67.0, 78.0)	70.0 (64.0, 79.0)	0.086
**Gender (Male) *n* (%)**	**170 (79.8)**	**154 (88.0)**	**0.027**
**BMI (kg/m^2^)**	**25.7 (22.8, 32.1)**	**27.6 (24.1, 32.4)**	**0.054**
**Smoking (current/ex)**	86 (40.4)/127 (59.6)	67 (38.3)/108 (61.7)	0.675
**Pack-years**	70.0 (50.0, 81.2)	60.0 (45.0, 100.0)	0.965
**Borg dyspnea scale upon admission**	**7.0 (5.0, 8.0)**	**5.0 (3.0, 6.0)**	**<0.001**
**Charlson comorbidity index score**	**2.0 (2.0, 4.0)**	**2.0 (1.0, 3.0)**	**0.004**
**pH upon admission**	**7.40 (7.36, 7.44)**	**7.42 (7.38, 7.45)**	**0.008**
**PO_2_/FiO_2_ upon admission**	**238.1 (189.9, 278.6)**	**276.2 (238.2, 305.6)**	**<0.001**
**PCO_2_ upon admission**	**48.2 (39.6, 59.0)**	**40 (34.9, 47.9)**	**<0.001**
**Duration of hospital stay (days)**	**10.0 (8.0, 14.)**	**5.0 (3.0, 7.0)**	**<0.001**
**CRP upon admission (mg/L)**	**7.3 (3.1, 19.9)**	**3.4 (0.7, 7.8)**	**<0.001**
**Need for NIV**	**61 (28.6)**	**30 (17.1)**	**0.008**

Data are presented as median (IQR unless otherwise indicated). Abbreviations: BMI: body mass index; PO_2_: partial oxygen pressure; FiO_2_: fraction of inhaled oxygen; PCO_2_: partial carbon dioxide pressure; CRP: C-reactive protein; NIV: non-invasive ventilation. Bold indicates statistically significant differences.

**Table 2 jpm-13-00686-t002:** Correlations between blood eosinophils and several patient characteristics.

Variable	*p*	r
**BMI (Kg/m^2^)**	**0.045**	**0.107**
**Arterial pH upon admission**	**0.030**	**0.121**
**Arterial PO_2_/FiO_2_**	**<0.001**	**0.248**
**Duration of hospital stay (days)**	**<0.001**	**−0.589**
**Dyspnea upon admission (Borg scale)**	**<0.001**	**−0.370**
**CRP (mg/dL)**	**<0.001**	**−0.334**

Abbreviations: BMI: body mass index; PO_2_: partial oxygen pressure; FiO_2_: fraction of inhaled oxygen; CRP: C-reactive protein; NIV: non-invasive ventilation. Bold indicates significant correlations.

**Table 3 jpm-13-00686-t003:** Multinomial logistic regression analysis for the prediction of the need of NIV in COPD subjects admitted to hospital for COPD exacerbation.

					95% C.I. for EXP(B)	
	*Β*	S.E.	*p*-Value	Exp(B)	Lower	Upper
Age	−0.036	0.017	0.028	0.964	0.934	0.996
Gender (female)	0.260	0.516	0.614	1.297	0.472	3.564
BMI	0.009	0.007	0.164	1.009	0.996	1.022
CRP	0.003	0.023	0.900	1.003	0.958	1.050
Charlson	0.157	0.122	0.199	1.170	0.921	1.486
Blood eosinophils < 150	1.317	0.327	<0.001	3.731	1.996	7.081

Abbreviations: BMI: body mass index; CRP: C-reactive protein; C.I.: confidence interval.

## Data Availability

Data is unavailable due to privacy or ethical restrictions.

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
