# Peer review of "Eosinopenia as a Prognostic Biomarker for Noninvasive Ventilation Use in COPD Exacerbations"

_jpm, 2023, doi:10.3390/jpm13040686_

Round 1
Reviewer 1 Report
This is a very interesting study about e the effectiveness of blood eosinophils for predicting the need NIV in subjects with COPD exacerbation. Eosinophil count from the first complete blood count was used to designate the eosinophil group, while the relationship between clinical characteristics and blood eosinophil counts was evaluated. Results showed that subjects with blood eosinophil number <150k/μL had a more severe disease on admission compared to subjects with 26 ≥150k/μL and required a longer hospital stay. Blood eosinophil number <150k/μL was an independent predictor of the use of NIV during hospital stay, while authors showed that during COPD exacerbation, low blood eosinophil levels on admission are related to more severe disease and can be used as a predictor of the need of NIV.
Author Response
Reviewer 1
This is a very interesting study about the effectiveness of blood eosinophils for predicting the need NIV in subjects with COPD exacerbation. Eosinophil count from the first complete blood count was used to designate the eosinophil group, while the relationship between clinical characteristics and blood eosinophil counts was evaluated. Results showed that subjects with blood eosinophil number <150k/μL had a more severe disease on admission compared to subjects with 26 ≥150k/μL and required a longer hospital stay. Blood eosinophil number <150k/μL was an independent predictor of the use of NIV during hospital stay, while authors showed that during COPD exacerbation, low blood eosinophil levels on admission are related to more severe disease and can be used as a predictor of the need of NIV.
We thank the reviewer for his/her comments
Reviewer 2 Report
The short report under evaluation relates to eosinopenia as an independent marker for the need for non-invasive ventilation in patients with acute exacerbation of COPD. My objections to the study are as follows:
1. Regarding the number of eosinophils, it should be clarified whether the count was automatic or subsequently revised.
2. Regarding the main message of the article, it should be clarified what was the reason for requiring non-invasive ventilation in the 91 patients listed in table 1. In other words, how many required it because of respiratory acidosis and how many because of hypoxaemic respiratory failure. In other words, the authors must demonstrate that the group of patients with eosinopenia strictly met criteria for non-invasive ventilation as specified in the guidelines. It is not sufficient to state that they were more frequently ventilated because there may be important selection biases, especially if the study was conducted in different healthcare centres.
3. The pH value, even if it reaches statistical significance, is normal in both groups, therefore, it cannot be stated that patients with eosinopenia have worse acid-base balance. Instead of this parameter, specify the patients who had respiratory acidosis in both groups.
4. PaCO2 values should be listed in table 1.
Author Response
Reviewer 2
The short report under evaluation relates to eosinopenia as an independent marker for the need for non-invasive ventilation in patients with acute exacerbation of COPD. My objections to the study are as follows:
- Regarding the number of eosinophils, it should be clarified whether the count was automatic or subsequently revised.
We thank the reviewer for his/her comment.
In the study we have used the number of blood eosinophils which was automatically provided by the analyzer providing the complete blood count results in the emergency department and before administration of any treatment to the patient.
A phrase has been included in the methods section as follows:
“The eosinophil count from the first complete blood count obtained in hospital (usually in the emergency department) was recorded and was used to determine the number of blood eosinophils. Blood sample collection and measurement of blood eosinophils was performed upon admission and before administration of any treatment.”
- Regarding the main message of the article, it should be clarified what was the reason for requiring non-invasive ventilation in the 91 patients listed in table 1. In other words, how many required it because of respiratory acidosis and how many because of hypoxaemic respiratory failure. In other words, the authors must demonstrate that the group of patients with eosinopenia strictly met criteria for non-invasive ventilation as specified in the guidelines. It is not sufficient to state that they were more frequently ventilated because there may be important selection biases, especially if the study was conducted in different healthcare centers.
We thank the reviewer for his/her comment
In our study patients received NIV due to respiratory acidosis. This is now clarified to the methods section as follows:
“Finally, all patients who underwent treatment with NIV suffered from respiratory acidosis.”
- The pH value, even if it reaches statistical significance, is normal in both groups, therefore, it cannot be stated that patients with eosinopenia have worse acid-base balance. Instead of this parameter, specify the patients who had respiratory acidosis in both groups.
We thank the reviewer for his/her comment. Indeed, pH median value and IQR were normal in both groups although there is statistically significant difference between values. However, these values are on admission. The aim of the study was to determine the prognostic value of blood eosinophils on the need of the requirement of NIV during hospital stay and not necessarily upon admission. This fact makes our observation even more important since it helps the physician to recognize early patients who are at risk for a poorer outcome and offer them a more intensive treatment and closer monitoring.
A comment has been added in the discussion as follows:
“Interestingly, in our study, the median pH value and interquartile range (IQR) were normal in both groups although there is a statistically significant difference between the two groups. However, we must point that these values were recorded on admission and the aim of our study was to determine the prognostic value of blood eosinophils on the need of the requirement of NIV during hospital stay and not upon admission. This fact makes our observation even more important since it helps the physician to recognize early those patients who are at greater risk for a poorer outcome and offer them a more intensive treatment and closer monitoring”.
- PaCO2 values should be listed in table 1.
We thank the reviewer for his/her comment.
PaCO2 values have been added in table 1 and now it appears as follows:
|
Variable |
Blood Eosinophils <150 cells/μL N=213 |
Blood Eosinophils ≥150 cells/μL N=175 |
p-value |
|
Age (years) |
73.0 (67.0, 78.0) |
70.0 (64.0, 79.0) |
0.086 |
|
Gender (Male) N (%) |
170 (79.8) |
154 (88.0) |
0.027 |
|
BMI (kg/m2) |
25.7 (22.8, 32.1) |
27.6 (24.1, 32.4) |
0.054 |
|
Smoking (current/ex) |
86 (40.4)/127 (59.6) |
67 (38.3)/108 (61.7) |
0.675 |
|
Pack-years |
70.0 (50.0, 81.2) |
60.0 (45.0, 100.0) |
0.965 |
|
Borg Dyspnea scale on admission |
7.0 (5.0, 8.0) |
5.0 (3.0, 6.0) |
<0.001 |
|
Charlson comorbidity Index score |
2.0 (2.0, 4.0) |
2.0 (1.0, 3.0) |
0.004 |
|
pH on admission |
7.40 (7.36, 7.44) |
7.42 (7.38,7.45) |
0.008 |
|
PO2/FiO2 on admission |
238.1 (189.9, 278.6) |
276.2 (238.2, 305.6) |
<0.001 |
|
PCO2 on admission |
48.2 (39.6, 59.0) |
40 (34.9, 47.9) |
<0.001 |
|
Duration of hospital stay (days) |
10.0 (8.0, 14.) |
5.0 (3.0, 7.0) |
<0.001 |
|
CRP on admission (mg/L) |
7.3 (3.1,19.9) |
3.4 (0.7, 7.8) |
<0.001 |
|
Need for NIV |
61 (28.6) |
30 (17.1) |
0.008 |